# Accuracy of Increment Core Method for Measuring Basic Wood Density and Moisture Content in Three *Catalpa* Species

**DOI:** 10.3390/plants14152339

**Published:** 2025-07-29

**Authors:** Xiping Zhao, Dongfang Wang, Pingping Guo, Qi Feng, Yuanping Deng

**Affiliations:** 1College of Horticulture and Plant Protection, Henan University of Science and Technology, Luoyang 471023, China; 230320191164@stu.haust.edu.cn (D.W.); 9903545@haust.edu.cn (P.G.); 9905183@haust.edu.cn (Q.F.); 2Library, Henan University of Science and Technology, Luoyang 471023, China; dypcdx@haust.edu.cn

**Keywords:** *Catalpa* spp., increment core, wood density, moisture content, test accuracy

## Abstract

Assessing wood moisture and density is essential to understanding ecological processes such as tree growth and wood formation. This study compared basic density and moisture content estimates for three *Catalpa* species (*Catalpa ovata* G. Don, *Catalpa bungei* C. A. Mey, and *Catalpa fargesii* Bureau) using three sampling methods (incremental cores, wood chips, and standard wood blocks). While strong correlations (r^2^ ≥ 0.99) were observed among all methods, the incremental core approach exhibited significant species-specific biases—overestimating density by 27.31–12.31% on average while underestimating moisture content by 5.61–30.51%. Despite its cost-effectiveness and minimal sample collection requirements, the method’s systematic deviations limit its applicability to multiple tree species. Consequently, we recommend developing species-specific linear calibration models that incorporate baseline data from standard wood block measurements to substantially improve estimation accuracy. This approach offers a practical, theory-supported solution for optimizing field sampling strategies in ecological research.

## 1. Introduction

*Catalpa* spp. wood has become increasingly valuable in modern wood product industries. This versatile timber is used for multiple purposes, including high-end furniture manufacturing, intricate artistic carvings, durable fence post construction, and premium coffin production [1,2]. The wood’s growing market demand stems from several distinctive characteristics: its attractive natural color palette, visually appealing decorative grain patterns, exceptional mechanical strength properties, and remarkable long-term durability [2,3,4,5,6]. Geographically, *Catalpa* spp. demonstrate impressive adaptability across diverse climates. These trees naturally thrive in temperate zones spanning both the Americas and East Asia [5,7,8]. Notably, China has made significant progress in cultivating fast-growing *Catalpa* spp. varieties to address increasing domestic timber requirements [9]. This cultivation initiative serves two purposes: meeting economic demands while supporting sustainable forestry practices.

However, a critical technical challenge persists in forestry research. Accurately assessing wood properties—particularly moisture content and basic density—remains essential for monitoring tree growth dynamics and evaluating forest health status [3,10]. Current measurement methodologies present notable limitations. While traditional destructive sampling techniques (such as extracting complete wood blocks) yield laboratory-grade accuracy [3,10], they are impractical for large-scale or long-term ecological monitoring programs. The inherently destructive nature of these methods prevents repeated measurements on the same trees, creating substantial barriers to sustainable forest management initiatives.

In temperate and cold-temperate forest ecosystems, increment borers have emerged as indispensable research instruments [11,12,13,14,15]. These specialized tools extract small-diameter wood cores from living trees, providing a relatively non-destructive sampling alternative. The technique’s widespread adoption results from several key advantages: relatively low equipment costs, minimal physical damage to trees, and efficient sample collection [16].

Despite these benefits, questions remain regarding the accuracy of increment cores in determining wood density and moisture content. Several factors contribute to potential discrepancies: core compaction during the coring process; material shrinkage during sample drying; anatomical variations between different tree species [17]. Empirical studies reveal inconsistent findings across species. Wahlgren and Fassnacht [18] documented that superior accuracy was obtained with loblolly pine cores compared to three other southern yellow pine species in their investigation. Gilmore et al. [19], examining similar species but from plantation sources, reported significantly reduced measurement errors compared to the findings reported by Wahlgren and Fassnacht [18]. These conflicting results emphasize an important principle: wood measurement accuracy exhibits pronounced species-specific variation. The physical properties of wood—particularly density and moisture retention—depend heavily on both botanical species and specific growth conditions [17]. Consequently, any application of increment cores for property measurement necessitates preliminary validation specific to the target tree species.

Currently, in addition to conventional methods such as drying and weighing [20], wood moisture and density of tree cores are also determined using computed tomography (CT) scanning or X-ray methods [21]. CT scanning is a relatively expensive method for measuring the distribution of moisture content within wood. Although this method can obtain visual images for qualitative analysis of moisture content, it lacks practical application value due to slow measurement speed and expensive equipment. Most X-ray methods still measure the density of the object being tested first, and then calculate its moisture content, but there are some challenges in terms of equipment cost and operational complexity [22]. Regardless of which method is used to measure moisture and density in the collected tree cores, measurement results often deviate from standard methods. Therefore, correcting the systematic error of incremental core methods through simple mathematical models has become a scientific problem to be solved in this field [23,24]. The most promising correction method involves creating species-specific calibration models that account for characteristic compaction effects during coring, species-specific shrinkage coefficients, and unique moisture retention properties. Implementing these calibration models would enable researchers to convert raw core measurements into accurate estimates of actual wood properties [25].

This investigation focuses on three economically significant *Catalpa* species cultivated in China: *Catalpa ovata* G. Don, *Catalpa bungei* C. A. Mey, and *Catalpa fargesii* Bureau. This research employs a comparative experimental design, evaluating three distinct measurement methodologies: increment core method (field sampling technique), wood chip method (rapid assessment approach), and wood block method (standard laboratory procedure). Key research objectives include establishing the measurement validity range of the increment core method, identifying systematic error generation mechanisms, and developing practical correction algorithms for *Catalpa* spp. By establishing calibrated measurement protocols for China’s major *Catalpa* species, this research will promote more efficient and sustainable forest management practices. The findings will support both ecological conservation objectives and timber industry development goals, demonstrating how scientific innovation can facilitate balanced resource utilization.

## 2. Results and Discussion

The results are summarized in Table 1. The average moisture contents of *Catalpa* spp. obtained in this study are similar to those reported by Guo et al. [10]; average wood density is higher than the results reported by Ya et al. [26], Xiao et al. [27], and Zhao et al. [28]. Mean values of moisture content and wood density were the same for ring segments and chips but significantly different from wood blocks (standard specimens) (*p* < 0.05). Measurements of ring segments and chips underestimated moisture content and overestimated wood density. The lower moisture content value obtained with ring segments and chips can generally be attributed to the greater moisture loss of smaller samples during collection and storage. For the moisture content and density of the wood, the coefficient of variation of the ring segments and chips are close to and greater than that of wood blocks. This is because smaller samples are prone to large relative errors during weighing, drying, and other operations, which can affect the accuracy of measuring wood density and moisture content [29].

Our study reveals a strong linear relationship between ring segments, wood chips, and wood blocks with both moisture content and basic density measurements, with regression coefficients (R^2^) consistently exceeding 0.99 across all comparisons (Figure 1a–d). This demonstrates that wood properties can be reliably estimated from small-diameter samples, offering significant advantages for field measurements. A high correlation (R^2^ = 0.992) was observed between increment cores and wood blocks (Figure 1a,b), while the comparison between ring segments and wood chips showed even stronger agreement (R^2^ > 0.996,) with a remarkably low root mean square error (RMSE < 0.065) (Figure 1c,d). These findings confirm that tree core sampling provides an effective non-destructive method for wood density determination, as previously suggested by Tomczak, Tomczak, and Jelonek [20]. Notably, ring segments have demonstrated clear practical advantages over wood chips—they are easier to collect in the field, require simpler measurement procedures, and cause minimal damage to standing trees [16]. Low RMSE values across all comparisons indicate that while systematic differences exist between sampling methods, these variations remain predictable and correctable. This consistency in measurement accuracy across different sample types and wood properties suggests that species-specific calibration models could be used to effectively standardize increment core measurements, making them a viable alternative to destructive sampling methods for forest inventory and growth monitoring. The strong correlations observed in our study provide strong evidence that properly calibrated increment cores can yield reliable wood property estimates while offering significant practical benefits for sustainable forest management.

Figure 2 presents a detailed analysis of how moisture content and wood density correlate with the length of ring segments, revealing some interesting patterns in our measurements. While both parameters show a relationship with segment length, the correlation is notably stronger for wood density than for moisture content. This difference is likely due to the physical distortions that occur during the increment coring process, as previously noted in the literature [30]. The standard method for calculating ring segment volume, following the approach outlined by Pérez-Harguindeguy et al. [31] and using Equation (1) based on measured length and diameter, appears particularly sensitive to these distortions. Specifically, the coring process tends to compress the wood material, causing the measured length of segments to be systematically shorter than their actual dimensions. This length underestimation directly affects density calculations because density is inversely proportional to volume. However, our quantitative analysis shows these effects remain remarkably limited, with linear regression coefficients (R^2^) below 0.02 for both parameters. This extremely weak correlation demonstrates that while the coring process introduces some physical distortions, they do not significantly compromise the measurement accuracy of either moisture content or wood density when using properly calibrated methods. Importantly, these findings suggest that ring segments can indeed be divided by growth year for analysis without substantial loss of measurement precision, providing practical flexibility for field sampling and subsequent laboratory analyses. The minimal impact of segment length variations on measurement accuracy further validates our approach of using increment cores as a reliable method for assessing wood properties in *Catalpa* species.

The deviation between increment core measurements and wood block reference values shows a clear upward trend (Figure 3). This pattern holds true for both moisture content and basic density calculations. Several key factors contribute to this phenomenon:

First, the porous structure of wood plays a critical role [32]. The inherent porosity creates variable water absorption during testing, directly impacting measurement accuracy. When measuring moisture content, this adsorption effect becomes particularly significant.

Second, the water displacement method used for green wood volume measurement introduces additional variables [33]. Although immersion involves quickly submerging the sample, uneven water absorption inevitably occurs. This absorption varies by wood species, specific gravity, and initial moisture level. As a result, density calculations often show higher deviations from reference values [34].

Third, the relationship between the deviation rate and measured values follows a consistent pattern: higher reference density means greater measurement deviation, while higher reference moisture means greater accuracy loss. This correlation suggests that dense wood samples absorb more water during testing, and moisture-rich samples exhibit greater dimensional instability.

Fourth, the compaction effect during coring amplifies these deviations. Mechanical stress from boring alters the wood’s cellular structure, particularly in latewood and high-density areas [35].

Finally, drying shrinkage further complicates density measurements [17]. The degree of shrinkage varies systematically with the initial moisture content and anatomical wood features.

These combined factors explain why deviation rates increase with increasing reference values. The effect appears more pronounced for density measurements than moisture content, although both parameters show a significant correlation (*p* < 0.05).

Our comparative analysis of moisture content and wood density measurements across three *Catalpa* species (*C. ovata*, *C. bungei*, and *C. fargesii*) reveals significant species-specific differences in measurement accuracy when using increment core sampling. As shown in Table 2, the deviation rates between ring segment measurements and wood block standards vary considerably with species, with *C. fargesii* demonstrating particularly pronounced differences compared to both *C. bungei* and *C. ovata*. This species-specific variation extends to wood density measurements as well, where *C. ovata* exhibits an average deviation rate nearly double that of the other two species. These findings strongly suggest that tree species significantly influence the accuracy of increment core sampling for both moisture content and basic density determination. The underlying reasons likely relate to fundamental anatomical differences between these *Catalpa* species, including variations in cell structure, vessel arrangement, and extractive content [2,26,36]. Such morphological differences affect how wood responds to the coring process, particularly regarding compaction during coring and shrinkage during drying, factors that directly impact both density and moisture measurements. Importantly, our results indicate that increment core sampling cannot be universally applied to different *Catalpa* species without considering these species-specific measurement biases. Observed discrepancies highlight the need for developing species-specific calibration models to correct core measurements, as the current one-size-fits-all approach appears inadequate for achieving precise wood property assessments in these three Chinese *Catalpa* species. This species-dependent variability in measurement accuracy has important implications for forest inventory practices and wood quality evaluation, particularly when managing mixed-species *Catalpa* plantations.

Figure 4 presents a comprehensive comparison of wood density and moisture content measurements obtained by ring segment sampling versus standard wood block method among three *Catalpa* species. The linear regression analysis without intercepts demonstrates remarkably strong correlations between the methods, with all regression equations showing R^2^ values exceeding 0.99 (Table 3). This exceptionally high correlation coefficient indicates that, despite species-specific variations, the fundamental relationship between ring segment and wood block measurements remains consistently strong for both parameters. However, when we introduced intercepts to account for systematic measurement biases between the increment core method and national standard methods, the results revealed more nuanced patterns. While the root mean square errors (RMSEs) showed minimal differences between the regression approaches, the coefficient of determination (R^2^) values dropped significantly, particularly for wood density measurements, where R^2^ fell below 0.3. This substantial decrease in explanatory power suggests that simple linear models without intercepts may mask important systematic differences between sampling methods. Interestingly, the moisture content measurements exhibited better model performance, maintaining R^2^ values above 0.5 for all species, with *C. fargesii* showing particularly strong agreement (R^2^ = 0.842). These findings highlight the complex relationship between sampling methods and wood properties, where density measurements appear more susceptible to methodological biases than moisture content assessments. The validation of our proposed calibration models produced particularly encouraging results, with all validation models demonstrating R^2^ values above 0.99 and relative root mean square errors (RRMSEs) below 0.05 (Figure 5). This exceptional predictive performance confirms that while direct measurements from increment cores may contain systematic errors, these can be effectively corrected through species-specific calibration models. The validation results provide compelling evidence that tree core sampling, when properly calibrated, offers a practical and accurate alternative for assessing wood moisture content, with potential applications in sustainable forest management and quality control for *Catalpa* wood products. Species-specific differences in model performance further underscore the importance of developing tailored calibration approaches rather than applying universal correction factors.

## 3. Materials and Methods

### 3.1. Sample Preparations

A total of 72 trees representing *C. ovata*, *C. bungei*, and *C. fargesii* were randomly selected for this study from a plantation located in Yanshi District (36°41′ N, 140°41′ W, elevation 60 m), Luoyang City, Henan Province, China. The study site features a continental climate characterized by cold, dry winters and hot, humid summers, with an annual mean temperature of 13.7 °C, average precipitation of 550 mm, and a frost-free period of approximately 210 days [37]. Key characteristics of the sampled trees are summarized in Table 4.

Prior to felling, a single increment core (5.15 mm diameter) was extracted from each tree at breast height using a standard increment borer (CO500, Haglöf Sweden AB, Långsele, Sweden). Immediately after extraction, each core was carefully sealed in a labeled plastic tube to prevent contamination and moisture exchange. The felling process was carefully planned to ensure that the increment core position would be avoided when cutting cross-sectional discs. Following tree felling, two thick discs (approximately 25 mm) were removed from breast height—designated Disc A and Disc B—positioned strategically to maintain sufficient distance from the core extraction site (Figure 6). To preserve sample integrity during storage, all core and disc samples were promptly frozen using dry ice (solid CO_2_), which effectively minimized water loss through sublimation while simultaneously slowing fungal growth and microbial activity. This careful sample handling protocol ensured that the physical and chemical properties of the wood material remained stable between field collection and laboratory analysis.

### 3.2. Determination of Basic Wood Density and Moisture Content

Wood blocks (20 mm × 20 mm × 20 mm), ring chips, and ring segments were sequentially prepared from Disc A, Disc B, and the increment core, respectively, progressing from bark to pith. Immediately after cutting, the fresh weight of each sample was precisely measured using an electronic microbalance with 0.0001 g sensitivity.

The green volume of each wood block or chip sample was determined using the water displacement method following Chinese Standard GB/T 1927.5-2021 [33]. Prior to submersion, samples were coated with beeswax to prevent water absorption, as wood’s hygroscopic and porous nature could otherwise affect measurement accuracy. The volume was then measured by recording the water displacement before and after complete immersion of the sample.

The green length (L) of each ring segment was measured using an electronic Vernier caliper (Nanjing Hailian Tools Co., Ltd., Nanjing, China) with 0.01 mm precision following the protocol established by Pérez-Harguindeguy et al. [31]. Sample diameter (D) was fixed at 5.15 mm as specified by the manufacturer. Volume calculations for each segment were performed using Equation (1) based on the measured dimensions. To minimize moisture exchange, weighing bottles were employed for these small ring chips and segment samples. All measurements were conducted in a temperature- and humidity-controlled cold storage environment maintained at 5 °C and 85% relative humidity.(1)$$V_{i}=\pi \times {\left(\frac{D}{2}\right)}^{2}\times L_{i}$$
where *V_i_* is the ring segment volume (mm^3^), *D* is the increment core diameter (5.15 mm), and *L_i_* is the ring segment length (mm).

Following Chinese Standard GB/T 1927.4-2021 [38], all wood block, ring chip, and ring segment samples were oven-dried at 103 ± 2 °C until reaching a constant weight. The dried samples were then promptly transferred to a desiccant-filled container and allowed to be equilibrated to room temperature in a controlled chamber prior to final weight measurement.

The moisture content was calculated differently for each sample type, with the ring segment moisture content (SMC) determined by dividing the weight difference between green and oven-dried states by the initial green weight of the segment. The wood block moisture content (BMC) followed the same calculation principle, using the green and oven-dried weights of the entire block. For ring chips, the moisture content (CMC) was calculated by comparing their green and dried weights relative to their initial green weight.

Wood density calculation methodology was consistent for all sample types, with each value representing the ratio of oven-dried weight to green volume. Specifically, the ring segment density (SWD) used the segment’s green volume, while the wood block density (BWD) relied on the block’s original dimensions. Likewise, the ring chip density (CWD) was derived from the chip’s green volume, ensuring uniformity in the measurement approach despite differences in sample preparation.

### 3.3. Statistical Analyses

Descriptive statistics including minimum, maximum, mean, standard deviation, and coefficient of variation were employed to summarize basic wood density and moisture content data. Duncan’s multiple comparison tests were conducted to evaluate the effects of different sampling methods on these wood properties. Linear regression analyses were performed to examine the relationships between moisture content and density measurements of wood blocks, ring chips, and ring segments, as well as the correlation between ring segment measurements and segment length, plus the deviation rates between ring segment and wood block measurements. To address systematic measurement errors, we developed dynamic calibration models as practical correction methods for species-specific density and moisture content determination. For model construction, 54 trees were used for calibration, while 18 samples were used for validation, with careful data screening to ensure comparable means and ranges between datasets. The intercept of a linear equation represents the boundary values of the constraint conditions [24]. In our study, the comparison of linear equations with and without intercepts is essential to evaluate the fit of models used to predict wood density and moisture content. Model performance was assessed using R^2^ (coefficient of determination), RMSE (root mean square error), and RRMSE (relative RMSE) metrics. All statistical analyses were conducted using IBM SPSS software (version 22.0, International Business Machines Corporation, Armonk, NY, USA), with statistical significance set at *p* < 0.05 determined by F-test.

The deviation rate (*D_r_*) was evaluated according to Equation (2) given below:(2)$$D_{r}=\frac{Y_{b}-Y_{s}}{Y_{b}}\times 100$$
where *Y_b_* and *Y_s_* indicate basic wood density or moisture content measured using wood blocks and ring segments.

## 4. Conclusions

This study evaluated the effectiveness of using growth increment cores to estimate wood moisture content and density in three *Catalpa* species. While the method proved unreliable for accurately determining wood density for all three species, it showed promise for assessing moisture content, albeit with notable species-specific variations. Direct core measurements exhibited a maximum moisture content deviation of 30.5%, with *C. bungei* and *C. ovata* displaying similar average deviations of approximately 11.5%, while *C. fargesii* showed a higher deviation rate of about 14%. These findings highlight the need for species-specific calibration. To enhance measurement accuracy, we recommend developing linear correction models based on standard methods for each species, which would significantly improve data reliability for both wood science research and practical forestry applications.

## Figures and Tables

**Figure 1 plants-14-02339-f001:**
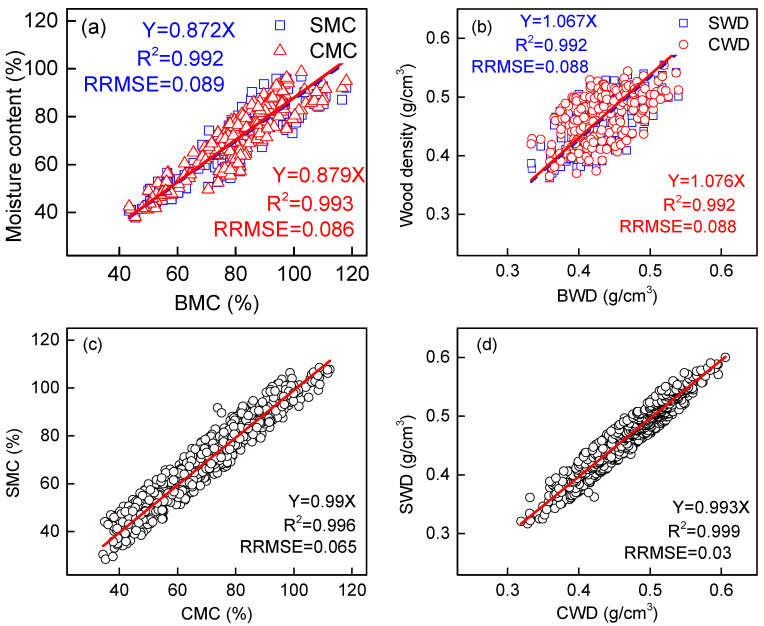
Relationship between moisture content and wood density measured using wood blocks, ring chips, and ring segments. The triangle, squares, and circles denote the measured values. The dashed blue lines and solid red lines denote the fitted values using linear regression without intercept and with 95% confidence interval. SMC, moisture content of ring segment; BMC, moisture content of wood block; CMC, moisture content of ring chip; SWD, wood density of ring segment; BWD, wood density of wood block; CWD, wood density of ring chip. (**a**) Relationship between SMC and BMC, as well as between CMC and BMC; (**b**) Relationship between SWD and BWD, as well as between CWD and BWD; (**c**) Relationship between SMC and CMC; (**d**) Relationship between SWD and CMD.

**Figure 2 plants-14-02339-f002:**
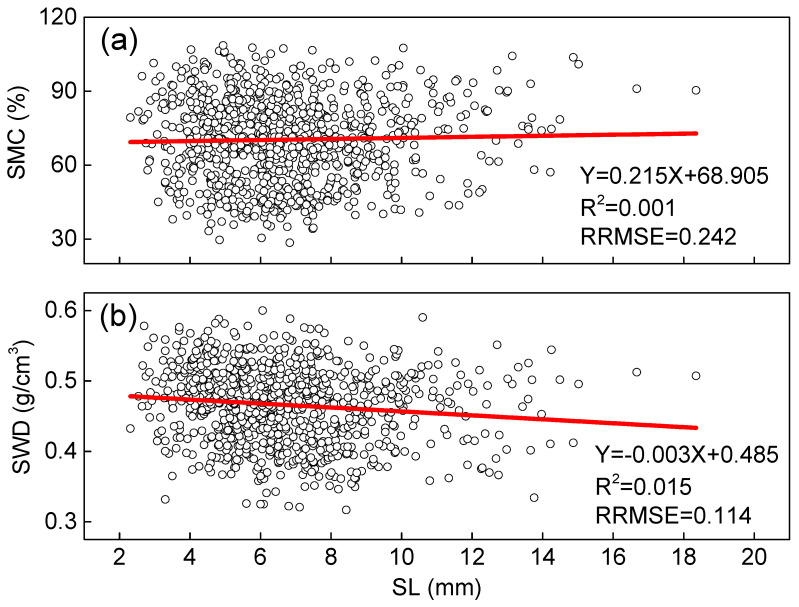
Moisture content (SMC) and wood density (SWD) as a function of length (SL) of ring segments. The circles denote the observed values. The solid lines denote the calculated values using unitary linear regression models with intercept (95% confidence interval). (**a**) Relationship between SMC and SL; (**b**) Relationship between SWD and SL.

**Figure 3 plants-14-02339-f003:**
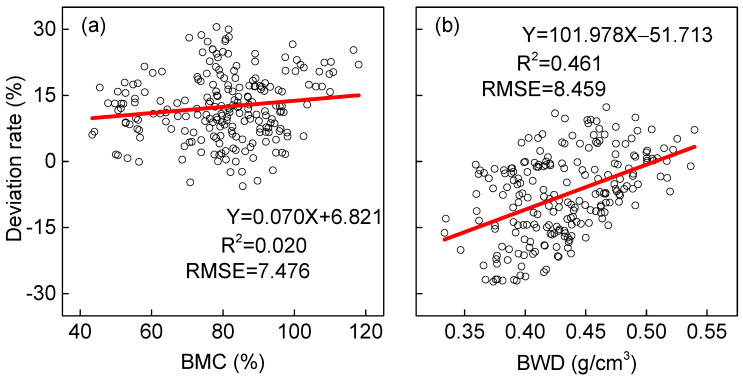
Deviation rate of moisture content (**a**) and wood density (**b**) measured using ring segments and wood blocks. The circles denote the observed values. The solid lines denote the calculated values using unitary linear regression models with intercept (95% confidence interval). BMC, moisture content of wood block; BWD, wood density of wood block.

**Figure 4 plants-14-02339-f004:**
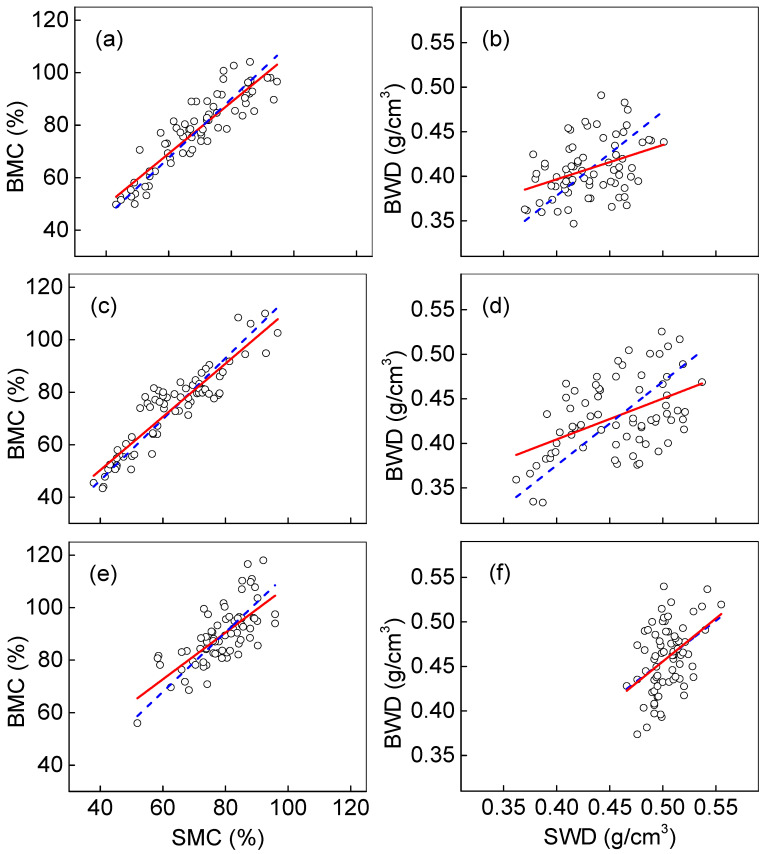
Relationship between values measured using wood blocks and values predicted by ring segments using calibration models for *Catalpa bungei* (**a**,**b**), *Catalpa fargesii* (**c**,**d**), and *Catalpa ovata* (**e**,**f**). The circles denote the measured values. The dashed blue lines and solid red lines denote the fitted values using linear regression with and without intercept, respectively. SMC, moisture content of ring segment; BMC, moisture content of wood block; BWD, wood density of wood block; SWD, wood density of ring segment. Fitted linear models with 95% confidence interval are shown in Table 3.

**Figure 5 plants-14-02339-f005:**
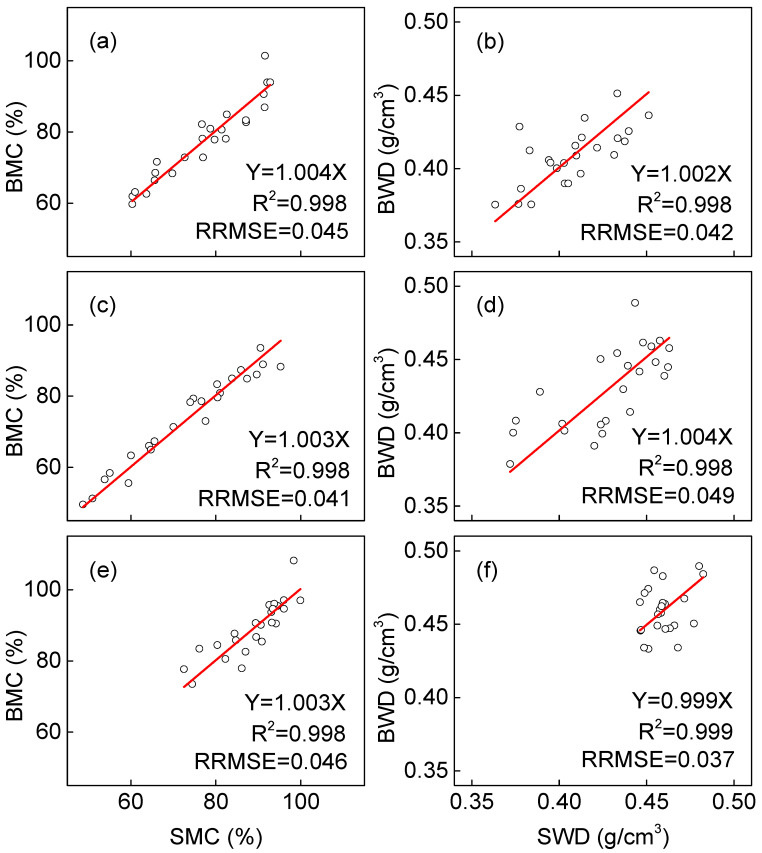
Relationship between values measured using wood blocks and values predicted by ring segments using validation models for *Catalpa bungei* (**a**,**b**), *Catalpa fargesii* (**c**,**d**), and *Catalpa ovata* (**e**,**f**). The circles denote the measured values. The solid red lines denote the predicted values. SMC, moisture content of ring segment; BMC, moisture content of wood block; BWD, wood density of wood block; SWD, wood density of ring segment. (For interpretation of the color references in this figure legend, the reader is referred to the web version of this article).

**Figure 6 plants-14-02339-f006:**
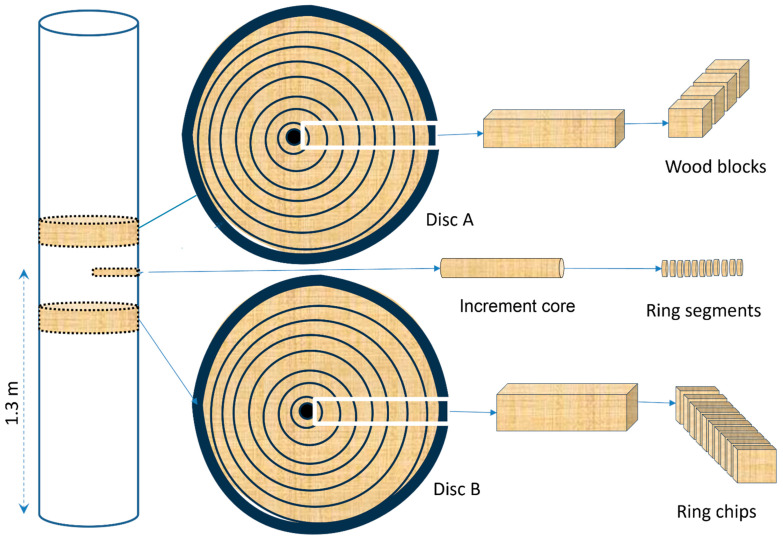
Tree sampling scheme.

**Table 1 plants-14-02339-t001:** Wood density and moisture content for ring segments, ring chips, and wood blocks.

Variable	Sample Shape	Number	Minimum	Maximum	Average	S.D.	C.V.
MC/%	Ring segment	895	28.55	108.56	70.35 b	16.99	0.24
	Ring chip	894	34.40	112.38	70.89 b	17.09	0.24
	Wood block	216	43.38	118.03	80.21 a	15.29	0.19
WD/(g∙cm^−3^)	Ring segment	895	0.32	0.60	0.47 a	0.054	0.12
	Ring chip	894	0.32	0.61	0.47 a	0.05	0.11
	Wood block	216	0.33	0.54	0.43 b	0.04	0.10

Note: Different lowercase letters indicate a significant difference among tree species based on the Duncan test (*p* < 0.05).

**Table 2 plants-14-02339-t002:** Deviation rate comparison of moisture content and wood density among tree species.

Tree Species	Deviation Rate	
	Moisture Content	Wood Density
*Catalpa bungei*	11.402 ± 0.829 b	−5.874 ± 1.047 b
*Catalpa fargesii*	14.286 ± 0.862 a	−6.6 ± 1.363 b
*Catalpa ovata*	11.501 ± 0.513 b	−10.316 ± 0.975 a

Note: Different lowercase letters indicate a significant difference among tree species based on the Duncan test (*p* < 0.05).

**Table 3 plants-14-02339-t003:** The linear regressions of wood density and moisture content with and without intercept established using wood block and ring segment measurements (*p* < 0.05).

Variable	Tree Species	Linear Regression Equation	R^2^	RRMSE
MC	*Catalpa bungei*	BMC = 1.125 × SMC	0.993	0.083
		BMC = 0.978 × SMC + 10.480	0.821	0.080
	*Catalpa fargesii*	BMC = 1.162 × SMC	0.993	0.088
		BMC = 1.014 × SMC + 9.803	0.842	0.084
	*Catalpa ovata*	BMC = 1.132 × SMC	0.992	0.091
		BMC = 0.888 × SMC + 19.429	0.512	0.089
WD	*Catalpa bungei*	BWD = 0.945 × SWD	0.993	0.083
		BWD = 0.386 × SWD + 0.242	0.143	0.074
	*Catalpa fargesii*	Y = 0.938 × SWD	0.990	0.10
		BWD = 0.458 × SWD + 0.221	0.226	0.088
	*Catalpa ovata*	Y = 0.912 × SWD	0.995	0.074
		BWD = 0.967 × SWD − 0.028	0.171	0.074

MC: moisture content; WD: wood density; SMC: moisture content of ring segment; BMC: moisture content of wood block; SWD: wood density of ring segment; BWD: wood density of wood block.

**Table 4 plants-14-02339-t004:** Basic conditions of sampled trees (mean ± std).

Species	Number	Diameter at Breast Height (cm)	Height (m)	Height of the Lowest Branch (m)
*Catalpa bungei*	24	11.6 ± 1.6	8.4 ± 0.4	2.8 ± 0.4
*Catalpa fargesii*	24	12.3 ± 6.6	8.3 ± 1.7	2.7 ± 0.2
*Catalpa ovata*	24	11.4 ± 2.2	12.8 ± 0.3	2.4 ± 0.1

## Data Availability

Data are contained within this article.

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
