# Peer review of "Accuracy of Increment Core Method for Measuring Basic Wood Density and Moisture Content in Three Catalpa Species"

_plants, 2025, doi:10.3390/plants14152339_

Round 1
Reviewer 1 Report
Comments and Suggestions for Authors
This manuscript provided a basis analysis of density and moisture content of three Catalpa wood species using three sampling methods. The testing methods are conventional. The novelty is the wood species and the slightly innovative sampling methods.
Specific comments
Although the authors stated one of the objectives as “propose a practical correction method to improve the accuracy of measuring wood density and moisture content using increment cores” (line 73-75), I failed to locate such correction method in the entire manuscript. What do you mean by “correction method”?
Line 163-165: Why is there a need to compared linear equations with intercepts with linear equation with no intercept? Was it used by other references?
Table 3. Instead of showing Y = mX, the linear regression equations should show BMC = a*SMC or BWB = b*SWD.
Besides the basic definitions provided in the nomenclature, BMC and BWD are not clearly defined in the Methods section and on the figures in the manuscript. How are they different from SMC and SWD? Please explain it explicitly.
Line 198: Please describe “drikold” as dry ice (solid CO2).
Line 251-252: “…three Catalpa spp. species. using growth increment cores was evaluated.” It is unnecessary to have “species” after spp.
Line 255: “It is almost affected by tree species.” Please elaborate and provide specific numbers.
Author Response
Dear Reviewer:
Thank you for your comments on our paper (Plants -3745018). We greatly admire your profound professional knowledge and rigorous scientific attitude. What is more valuable is that you not only pointed out the problems in our manuscript, but also provided valuable suggestions for modification. The main corrections with red text in the paper and responses to your comments are as follows:
- Although the authors stated one of the objectives as “propose a practical correction method to improve the accuracy of measuring wood density and moisture content using increment cores” (line 73-75), I failed to locate such correction method in the entire manuscript. What do you mean by “correction method”?
Response to (1): The meaning of "correction method" is to establish calibration models to reduce the systematic error of incremental core method. I'm sorry we missed this point. We explained it in the introduction, and emphasized it in the materials and methods.
- Line 163-165: Why is there a need to compared linear equations with intercepts with linear equation with no intercept? Was it used by other references?
Response to (2): The intercept of a linear equation may represent the boundary values of the constraint conditions. In our study, the comparison of linear equations with and without intercepts is essential to evaluate the fit of models used to predict wood density and moisture content. The intercept of the linear equation implies the baseline value at which the measurement results of the increment core method may deviate from those of the standard method. There is not much difference in root mean square error (RMSE) between the two types of linear fitting, the coefficient of determination (R2) is notably lower when an intercept is included, particularly for wood density. This suggests that the models without intercepts have a better fit. On the other hand, the fitting effect for moisture content is satisfactory, with R2 values above 0.5, and especially high for Catalpa fargesii trees, reaching 0.842. The validation of the proposed calibration model further confirms the high fitting degree between predicted and measured values, with all validation models having R2 values above 0.99 and relative root mean square error (RRMSE) less than 0.05. This indicates that using tree cores to measure wood moisture content is feasible and that established linear models can enhance measurement accuracy.
Linear equations were used by other references. For example, linear regression equations for prediction of whole-tree density from a core sample were established in Raymond 's study (Doi: 10.1007/s002260000078).
- Table 3. Instead of showing Y = mX, the linear regression equations should show BMC = a*SMC or BWB = b*SWD.
Response to (3): Based on your kind suggestion, we have changed the expression of variables in linear regression equations.
- Besides the basic definitions provided in the nomenclature, BMC and BWD are not clearly defined in the Methods section and on the figures in the manuscript. How are they different from SMC and SWD? Please explain it explicitly.
Response to (4): Based on your suggestion and combined with another Reviewer's comment, the abbreviations are defined in the Methods section and on the figures in the manuscript.
- Line 198: Please describe “drikold” as dry ice (solid CO2).
Response to (5): We are sorry for our less rigorous writing. Based on your kind suggestion, we have corrected the citation.
- Line 251-252: "…three Catalpa spp. species. using growth increment cores was evaluated. " It is unnecessary to have "species" after spp.
Response to (6): Thank you very much for your kind comment. At your suggestion, we deleted the word "species".
- Line 255: “It is almost affected by tree species. ” Please elaborate and provide specific numbers.
Response to (7): At your suggestion, we have supplemented the specific numbers of each tree species.
Reviewer 2 Report
Comments and Suggestions for Authors
The introduction is well-structured – it presents the importance of Catalpa spp. wood in various industries and highlights the need for effective, non-invasive methods for measuring wood density and moisture. The authors clearly identify a research gap and justify the study’s aim.
The methodology is described in detail. The researchers compared three wood sampling methods (increment cores, wood chips, and blocks) to measure moisture and density in three Catalpa species. The description of sample preparation, volume measurement (displacement method), drying procedures, and data analysis is thorough. 1) Lines 212-214: In my opinion, you can use the shorter form, Pérez-Harguindeguy et al. 2) It is widely known how wood moisture content is typically calculated, but I couldn't seem to find this information clearly described in the methodology section. If I have overlooked it, I would appreciate it if you could point out where exactly the procedure for determining moisture content is explained.
3) Line 100: chips not chaps
For example, in figure 1, you used some abbreviations, the explanation of which I cannot find in the text.
This article is a solid contribution to the field of non-invasive wood property measurement. The authors conducted a thorough comparative analysis of different wood sampling techniques, proposed species-specific calibration models, and offered practical recommendations for forestry applications. The manuscript only requires minor improvements in language and formatting.
Comments on the Quality of English Language
The text is grammatically correct, though at times the writing is awkward – likely due to translation from Chinese or writing in English as a second language. Some sentences are too long or complex. Scientific terminology is generally used correctly, but the manuscript would benefit from language editing prior to publication.
Author Response
Thank you for your insightful feedback on our manuscript (Plants -3745018). We sincerely appreciate your expertise and the thoughtful, constructive suggestions you provided, which have significantly strengthened our work. Below, we present the revised manuscript with key corrections highlighted in red, along with our point-by-point responses.
- Lines 212-214: In my opinion, you can use the shorter form, Pérez-Harguindeguy et al.
Response (1): At your suggestion, we have changed the citation form of the reference in the text.
- It is widely known how wood moisture content is typically calculated, but I couldn't seem to find this information clearly described in the methodology section. If I have overlooked it, I would appreciate it if you could point out where exactly the procedure for determining moisture content is explained.
Response to (2): We are sorry for our less rigorous writing. We highlight the description of the steps for measuring moisture content. The text highlighted in green. It should be noted that the measurement steps for moisture content are based on Chinese standards (GB/T 1927.4-2021), which we have overlooked. We have added it to the revised test.
- Line 100: chips not chaps.
Response to (3): We are sorry for our less rigorous writing. We have corrected the word.
- For example, in figure 1, you used some abbreviations, the explanation of which I cannot find in the text.
Response to (4): Based on your suggestion and combined with another Reviewer's comment, abbreviations are defined in the Methods section and on the figures in the manuscript.
- This article is a solid contribution to the field of non-invasive wood property measurement. The authors conducted a thorough comparative analysis of different wood sampling techniques, proposed species-specific calibration models, and offered practical recommendations for forestry applications. The manuscript only requires minor improvements in language and formatting.
Response to (5): Thank you very much for your kind comment. At your suggestion, we invited a native English speaker to improve the language of our manuscript.
Round 2
Reviewer 1 Report
Comments and Suggestions for Authors
The revision is mostly satisfactory.
One comment is:
Please include your response to my comment (2) on "Why is there a need to compared linear equations with intercepts with linear equation with no intercept? Was it used by other references?" into an appropriate location in the manuscript.
Author Response
Dear Reviewer:
Thank you for your valuable feedback on our manuscript (Plants -3745018) again. We have carefully addressed the comment and made the requested revision. Below is our point-by-point response to your suggestion.
Comment: Please include your response to my comment (2) on "Why is there a need to compared linear equations with intercepts with linear equation with no intercept? Was it used by other references?" into an appropriate location in the manuscript.
Response: Based on your kind suggestion, we have added our response your comment (2) into an appropriate location in the manuscript.
We hope the revised manuscript meets your expectation. Please let us know if further clarifications are needed.
Best regards,
Xiping Zhao
College of Horticulture and Plant Protection,
Henan University of Science and Technology,
263 Kaiyuan Avenue, Luoyang, P.R. China 471023